# Effect of Walnut Supplementation on Dietary Polyphenol Intake and Urinary Polyphenol Excretion in the Walnuts and Healthy Aging Study

**DOI:** 10.3390/nu15051253

**Published:** 2023-03-02

**Authors:** Rita I. Amen, Rawiwan Sirirat, Keiji Oda, Sujatha Rajaram, Ifeanyi Nwachukwu, Montserrat Cofan, Emilio Ros, Joan Sabate, Ella H. Haddad

**Affiliations:** 1Center for Nutrition, Healthy Lifestyle and Disease Prevention, School of Public Health, Loma Linda University, Loma Linda, CA 92350, USA; 2Lipid Clinic, Endocrinology and Nutrition Service, Institut d’Investigacions Biomèdiques August Pi Sunyer, Hospital Clínic, 08036 Barcelona, Spain; 3CIBER Fisiopatología de la Obesidad y Nutrición (CIBEROBN), Instituto de Salud Carlos III, 28029 Madrid, Spain

**Keywords:** polyphenols, dietary bioactive components, walnuts

## Abstract

Among all tree nuts, walnuts contain the highest total polyphenols by weight. This secondary data analysis examined the effect of daily walnut supplementation on the total dietary polyphenols and subclasses and the urinary excretion of total polyphenols in a free-living elderly population. In this 2-year prospective, randomized intervention trial (ID NCT01634841), the dietary polyphenol intake of participants who added walnuts daily to their diets at 15% of daily energy were compared to those in the control group that consumed a walnut-free diet. Dietary polyphenols and subclasses were estimated from 24 h dietary recalls. Phenolic estimates were derived from Phenol-Explorer database version 3.6. Participants in the walnut group compared to the control group had a higher intake of total polyphenols, flavonoids, flavanols, and phenolic acids in mg/d (IQR): 2480 (1955, 3145) vs. 1897 (1369, 2496); 56 (42,84) vs. 29 (15, 54); 174 (90, 298) vs. 140 (61, 277); and 368 (246, 569) vs. 242 (89, 398), respectively. There was a significant inverse association between dietary flavonoid intake and urine polyphenol excretion; less urinary excretion may imply that some of the polyphenols were eliminated via the gut. Nuts had a significant contribution to the total polyphenols in the diet, suggesting that a single food like walnuts added to habitual diet can increase the polyphenol intake in a Western population.

## 1. Introduction

It is well-recognized that walnuts have a favorable nutrient and fatty acid profile, and their consumption is effective in reducing blood lipids [1,2] and in modifying inflammation and endothelial dysfunction [3,4,5], thus reducing the risk of cardiovascular disease [2,6]. The risk lowering effects of walnuts as demonstrated by supplementing diets with the nuts, are greater than predicted based on the amount and nature of the fat consumed [7]. Evidence suggests that the phenolic phytochemicals found in walnuts and other nuts increase antioxidant defenses and reduce inflammation [7]. A recent review and meta-analysis summarizing the findings from several randomized controlled trials showed that incorporating polyphenol-rich foods impacts blood lipids by increasing high-density lipoprotein (HDL) and lowering low-density lipoprotein (LDL) [8]. Diets rich in polyphenols such as the Mediterranean diet, emphasizing olive oil and walnuts, report decreased blood lipids and inflammatory markers [6,9].

Walnuts are composed of an outer green husk with a hard shell inside the husk containing a walnut kernel covered with a seed coat or pellicle, where most polyphenols reside [10,11]. Among the nuts, walnuts contain the highest concentrations of polyphenols, averaging 2500 gallic acid equivalent (GAE) per 100 g [12]. Current comprehensive analyses of walnut polyphenols using chromatographic, and mass spectrometric techniques have identified hundreds of compounds in the walnut kernel including hydrolysable and condensed tannins, flavonoids, flavanols, phenolic acids, and lignans [13,14]. A study investigated the postprandial effect of walnut intake on the plasma total polyphenols and showed increased concentrations of plasma polyphenols 30 min following ingestion, which reached a peak at 90 min [1]. Therefore, walnuts may contribute to dietary polyphenols and may offer protective health benefits.

Several studies have recently explored dietary compositional changes produced by adding walnuts to the diet. In a randomized parallel design intervention, participants at risk for type 2 diabetes who added walnuts to their habitual diet increased their energy, protein, total fat, and magnesium intake [15] and had a non-significant decrease in sodium, empty calories, and dairy products [16]. In similar studies, individuals randomized to the walnut group showed higher intakes of protein, polyunsaturated fatty acids, both omega-3 and omega-6, but lower intakes of carbohydrates, animal protein and saturated fatty acids [17], and, in a cross-over study, participants who consumed walnuts additionally increased their dietary fiber, calcium, phosphorus, magnesium, and zinc intake [18]. However, no studies have as yet investigated whether the inclusion of walnuts in the diet influences the dietary intake of polyphenols or urinary polyphenol excretion. In this secondary data analyses of the Walnuts and Healthy Aging study (WAHA) [19], we investigated the impact of consuming walnuts on the dietary intake of total polyphenols and their sub-classes (flavonoids, flavanols, and lignans), and on the total urinary polyphenol excretion. Therefore, the aim of the current secondary analysis of data from the WAHA study was to determine whether long-term inclusion of walnuts (*Juglans regia* L.) in the daily diet increases polyphenol intake and the urinary excretion of phenolic metabolites. 

## 2. Materials and Methods

### 2.1. Study Design and Participants

The WAHA study was a 2-year parallel group, observer-blinded randomized controlled trial (RCT) examining the effect of the usual diet supplemented with walnuts at 15% (30–60 g/d) of energy compared to a walnut free habitual diet on the aging outcomes in elderly participants [17,20,21,22,23,24,25,26,27]. The parent study was a dual center clinical trial and was carried out in Barcelona, Spain and at Loma Linda University (LLU) in California, USA, from 2014 to 2016. However, only data collected from participants at LLU were used in the current secondary data analyses. The WAHA study was conducted in accordance with the guidelines of the Declaration of Helsinki and was approved by the Ethics Committee of the Loma Linda University Institutional Review Board (IRB 5120066). All participants provided their written informed consent before enrolment. The WAHA study clinical trial (NCT01634841) is registered at www.clinicaltrials.gov (accessed on 23 February 2023).

Detailed information about the WAHA study has been published elsewhere [19,26]. Briefly, candidates for this study were elderly ambulatory men and women aged 63–79 years. Exclusion criteria were inability to undergo neuropsychological testing; previously diagnosed neurodegenerative disease; prior stroke, significant head trauma, or brain surgery; relevant psychiatric illness; major depression; morbid obesity; uncontrolled diabetes; uncontrolled hypertension; prior chemotherapy; allergy to walnuts; habitual consumption of tree nuts (>2 servings/week); or customary use of fish oil, flaxseed oil, and/or soy lecithin. A total of 656 subjects were recruited and assessed for eligibility by the LLU team and 356 met the eligibility criteria and were randomized into the study. Of the total sample of randomized participants, 300 were selected for the current analysis, as shown in the flowchart in Figure 1. 

### 2.2. Sociodemographic, Anthropometric, and Biochemical Outcomes

Demographic data, anthropometric measurements, and dietary and lifestyle habits were collected from the participants at the baseline according to the study protocols [19]. Anthropometric measurements were carried out by trained professionals, and sociodemographic data and lifestyle habits were inputted using self-reported study questionnaires. Blood and spot urine samples were collected at the baseline and at the end of each year of intervention, aliquoted, and stored at −80 °C until analysis. All routine biochemical analyses and the determination of urinary creatinine concentration were performed at the completion of the study in the same laboratory to control for between-assay variability, as previously reported [20]. 

### 2.3. Estimation of Dietary Nutrient and Polyphenol Intake

Collection of dietary recall data and nutrient analysis was performed using the Nutrition Data System for Research (software version 2018) developed by the Nutrition Coordinating Center, University of Minnesota, Minneapolis, MN. The 24-h dietary recalls were obtained by telephone or face-to-face interviews using a multiple-pass approach to capture information about the food items, beverages, and dietary supplements consumed during the past 24 h and the nutrient estimates were acquired using the systems’ nutrient database. A total of five unannounced dietary recalls per participant were obtained at random times during the study, and these included at least one weekend day. The dietary intake data were collected by trained research dietitians and conducted at diverse intervals over the 2 year study duration to account for seasonal variations of food intake [17].

The polyphenol content of foods and beverages reported in the 24-h recalls were generated from the Phenol-Explorer database (version 3.6) [28]. The Phenol-Explorer database compiles the total polyphenol content of foods based on analyses performed using the Folin–Ciocalteu (F–C) reagent, whereas chromatography methods are used to estimate polyphenol subclasses. In the current study, the following variables were estimated based on data from Phenol-Explorer: the subclass total flavonoids consisted of flavones, flavonols plus anthocyanins, and the subclass phenolic acids were phenolics obtained either by chromatography or by chromatography after hydrolysis; the subclass flavanols were obtained by chromatography or normal phase HPLC; and the subclass lignans was obtained by chromatography after hydrolysis.

The contribution of food items to the total polyphenols, flavonoids, flavanols, phenolic acids, and lignans were entered into the dietary database in milligrams per 100 g per day. Food items found in the 24-h dietary recalls (24-HDR) and food composition data were matched and the intake of total dietary polyphenols and phenol subclasses was estimated using the following equation: 24-HDRs = Σ Pn × Gn. Here, p is the mg of phenolic compound per 100 g food, and G is the reported portion size of food in grams.

### 2.4. Urinary Total Polyphenols

Spot urine samples [29] were collected from the WAHA participants at the baseline and at the end of the first and second years of the study. Spot urine samples were collected in the morning at the time participants came in for their fasting blood draw but were not the first void. Samples were processed and stored at −80 °C until use. The total urinary polyphenol concentrations in the spot urine samples were determined using the modified rapid Folin–Ciocalteu (F–C) method, as previously described [30]. Briefly, following solid phase extraction for the removal of interfering substances using Oasis Max cartridges (Waters Corp. Milford, MA, USA), the samples were loaded on 96-well plates (Waters Corp., Milford, MA, USA) for testing using the Folin–Ciocalteu reagent. The Bio Tek Synergy HT spectrometer (Bio Tek, Winooski, VT, USA) was used to measure the resulting absorbance at 765 nm. All analyses were run in triplicate using gallic acid as the standard. Urine creatinine was determined using the Jaffe’ alkaline picrate microplate method as published [30].

### 2.5. Statistical Analyses

From the LLU cohort, a total of *n* = 356 subjects were randomized, but only 300 subjects were included in this secondary data analysis. A total of 34 subjects were excluded due to missing data. Dietary polyphenol variables of total polyphenols, total flavonoids, flavanols, phenolic acids, and lignans were energy-adjusted using the residual method and then averaged for each subject. Spot urine polyphenol concentrations in mg GAE/L were adjusted by creatinine concentration to account for urine dilution. Mann–Whitney tests were used for these variables for between-group comparisons. Means and standard deviations (SD) of polyphenol intake by food group were reported.

In the descriptive analysis of urinary polyphenols, the means (SD) by treatment and time were determined. To compare the morning spot urine polyphenol excretion between treatment groups, linear regression mixed models fitted for both variables (mg GAE/L, mg GAE/g Cr) included the treatment, time, treatment × time interaction, age, gender, and BMI as fixed-effects terms and the participants as a random-effects term. To examine the association between spot urine polyphenol excretion at year 2 and the dietary intake of polyphenols and subclasses, a linear regression model was fitted for each combination of urine polyphenol (dependent variable) and log dietary polyphenol (independent variable), while adjusting for age, gender, and BMI. All analyses were performed using R version 4.2.2 with a significance level at *p* < 0.05.

## 3. Results

The subject characteristics as observed in Table 1 show that there were more women than men enrolled in the study. In the walnut group, 63% were women and 37% were men, and in the control group, 68% were women and 32% were men. 

Table 2 describes the mean dietary intake of macronutrients by the treatment group over a 2-year period. A total of 1242 sessions were held to collect 24-h dietary recalls from the -participants. A total of five 24-h dietary recalls were collected from most participants (range 1–5 recalls) at random times during the duration of the study. The walnut group had a significantly higher energy intake, total dietary fiber, and total fat intake compared to the control group.

Table 3 describes the mean dietary intake of phenolics by treatment group for a 2 year period. Compared to the control group, participants in the walnut group had a significantly higher mean intake of total polyphenols, flavonoids, flavanols, and phenolic acids in mg/d. There were no significant differences in lignan intake. 

Table 4 shows the contribution of the various food groups to the daily intake of total polyphenols and phenolic subclasses by treatment group. Of the food groups consumed, the mean intake of nuts showed that walnuts significantly (*p* < 0.001) contributed to the total polyphenol intake in the walnut group in mg/d 632 (182) compared to the control 40 (7). Results of the polyphenol intake by food group also showed that nuts were a significant contributor to all other major subclasses including flavonoids (flavones, flavonols, and anthocyanidins, with the exception of lignan (*p* = 0.513) compared to the other food categories. 

Table 5 shows the comparison of urinary polyphenol excretion between the control and walnut groups at the baseline and at the end of years 1 and 2. Urinary polyphenols and creatinine were measured in the spot urine samples obtained from the participants at the same clinic visit when the fasting blood samples were drawn. From the baseline, the excretion of polyphenols in the walnut group in the first year approached significance at a 0.066 *p* value, but not in the second year or when the values were adjusted for urinary creatinine excretion. The values were similar at the baseline and years 1 and 2 in the control group. The results show that there were no significant differences between the intervention groups at any time point.

Table 6 describes the association between dietary and urinary polyphenols in year 2. Results of the linear models showed that there was a significantly negative association between the total urinary polyphenols and the log of total dietary flavonoids (*p* = 0.0316). There were no significant associations with any other dietary polyphenols.

## 4. Discussion

In this sub-study of the WAHA trial, we showed that the daily ingestion of walnuts for 2 years significantly increased the total dietary polyphenols and the subclasses of flavonoids, flavanols and phenolic acids in healthy elderly participants. To our knowledge, this is the first study to show that the inclusion of a single food (i.e., walnuts), with no other changes made to the usual diet, could significantly increase the total polyphenol intake. As expected, those who ate walnuts daily also showed higher intakes of energy, fiber, total fat, and unsaturated fatty acids. 

The results of this trial also show that participants in the walnut group consumed significantly higher amounts of total polyphenols and flavonoids (flavones, flavonols, and anthocyanidins), flavanols, and phenolic acids from nuts compared to those in the control group. This finding demonstrates that a single food such as walnuts can increase the intakes of total polyphenols and the polyphenol subclasses except for lignans. The walnut group had a higher intake of total polyphenols from fruits, and the flavanols and lignans from vegetables. The median daily intake of the total polyphenols of the control and walnut groups at 1897 mg/d and 2480 mg/d, respectively, of this elderly cohort residing in California was higher compared to that reported in adults in the U.S. by the National Health and Nutrition Examination Survey (NHANES) at 884 mg per 1000 kcal per day [31]. Like the current study, beverages such as tea, coffee, red wine, and fruit juices, vegetables, and fruits were the main contributors to the total polyphenols, flavonoids, and phenolic acids by NHANES [31] and through a recent systematic review of 91 studies from multiple countries [32]. A study [33] that examined the intake of dietary polyphenols by vegetarian status showed that a coffee intake, being a single food item, was the number one contributor to phenolic intake. Given that the WAHA intervention participants consumed walnuts at ~15% of their energy intake, the total polyphenol content of 2431.52 per 100 g walnuts would have added polyphenols to their diet [4].

While the walnut group showed a higher daily intake of polyphenols, this was not reflected in the urinary excretion of polyphenols tested in the spot urine samples obtained at the baseline, and at the end of either year 1 or year 2. Increased polyphenol metabolites have been identified in urine following the consumption of plant-based foods, suggesting that selected urinary polyphenols could be useful biomarkers to assess the intakes of polyphenol-rich foods and diets [34,35]. Most bioavailable dietary polyphenols have a relatively short half-life, estimated at 1 to 24 h following intake, and studies quantifying polyphenol biomarkers in urine have used 24-h urine collections following the consumption of test foods or diets [36,37,38]. Studies that have used 24-h urine samples were able to capture a wide range of polyphenols and positive relationships between dietary intake and urinary excretion of polyphenols [39,40]. One can conclude that the morning spot void used in our study could have resulted in poor collection of most polyphenols excreted in the urine over a 24-h period. However, similar studies found an increase in the concentration of phenolics in the spot morning urine following the intake of polyphenol-rich foods such fruits and vegetables [41,42]. Therefore, it is unclear why the daily inclusion of walnuts in the diet did not result in consistent increases in the concentration of polyphenols in our fasting spot urine samples collected following the first morning void. It is important to note that the rapid Folin–Ciocalteu (F–C) assay with solid-phase extraction optimized by Medina-Remón et al. [43] was used in this study to determine the total polyphenols in urine was validated in the spot urine samples collected from individuals consuming fruits, vegetables, tea, and red wine [44], and it has not been validated using walnuts.

In relation to the total polyphenol concentrations in the spot urine samples, our analyses did not show associations either with the dietary total polyphenols or flavanols and phenolic acid subclasses, but disclosed an inverse association with an intake of the flavonoid (flavanones + flavones) category. Studies in which the total urine polyphenols were measured with the F–C assay have shown weak to moderate associations with dietary polyphenol intakes in the 24-h [39], 12-h overnight [33], and morning urine sample collections [43]. In adults prescribed a high vegetable and fruit diet, the fasting spot urine samples collected after the first morning void and tested using liquid chromatography-mass spectroscopy disclosed an inconsistent association between the total urinary polyphenols and total polyphenol intake, while the linear mixed model analysis showed a non-significant inverse association between the total urine polyphenols and polyphenols from fruit [41]. The inverse correlation with fruit ingestion is consistent with our findings, since fruits are rich sources of the flavanone and flavone subclasses. It has been hypothesized that inverse correlations may be due to components of the food matrix that inhibit the intestinal absorption and urinary excretion of polyphenols [45,46]. The observed lack of associations or inverse associations may also be explained by the short half-life of bioavailable polyphenols and their metabolites, which may be absorbed and excreted within a short time period following intake [38,47]. Future studies should utilize 24-h urine collection following the ingestion of walnuts as the best method of capturing the majority of polyphenols. Additionally, reduced urinary excretion may imply that some of the polyphenols were eliminated via the gut, an effect likely to have a favorable impact on the intestinal microbiome [48,49,50,51,52,53].

The quality of phenolic compounds present in walnuts is diverse, ranging from simple phenolic acids and flavonoids to highly polymerized molecules such as tannins. Walnut phenolics are usually found at the highest concentration in the seed coat (also called the pellicle) surrounding the edible kernel and may be bound to other plant components such as carbohydrates and proteins. Consequently, some polyphenolic compounds might not be released in compositional testing studies. Walnuts are distinguished by the predominance of the hydrolysable gallotannins, glansrins (ellagitanins), and ellagic acid and the condensed tannins that are polymers of flavan-3-ol (flavanol) catechin and epicatechin subunits. The Polyphenol-Explorer 3.6 database reports an average amount of total polyphenol assayed by the F–C reagent as 1575 mg/100 g of kernel. Average amounts per 100 g of the total flavonoids, flavanols, and phenolic acids in walnuts are reported as 65 mg, 60 mg, and 449 mg, respectively [28,54,55,56,57].

Aside from the wide diversity and complexity of the phenolic substances found in walnuts, a number of other factors complicate efforts to obtain the exact accounts of their polyphenol composition. The concentration of phenolic compounds from different genomic walnut species and cultivars have been found to vary widely, with mean coefficients of variation of 25% or greater [58]. In addition, the climate, soil characteristics, agricultural practices, storage, and manipulation influence the phenolic content of the nuts [58,59]. Studies have shown substantial differences in the composition depending on the solvent or method (maceration, sonication) employed to extract phenols from walnuts [60]. The results are also influenced by whether the walnut kernel is raw, mildly heated or roasted, or whether it is defatted prior to extraction [4]. Current liquid chromatography techniques coupled with high resolution mass spectrometry and electrospray ionization tandem mass spectrometry have successfully been employed to identify and quantify phenolic compounds in walnuts found in soluble free, soluble esters, or conjugated and insoluble bound forms, thus providing more inclusive phenolic profiles than those reported by Phenol-Explorer [14,61]. 

It is important to note that the concentration of polyphenol in urine is determined by factors beyond the walnut phenol content and its structural matrix, but is related to human physiology, mainly sex and age, along with factors such as digestive and metabolic efficiency and the gut microbiota [62]. Depending on their structural complexity and solubility, it has been estimated that only 5–10% of the total dietary polyphenols reaching the small intestine are absorbed, with the maximum plasma concentrations attained at 30 min following ingestion. Urinary excretion generally peaks after about 8 h of intake [38,62,63]. Unabsorbed polyphenols reach the large intestine, where they undergo enzymatic action by the microbiota to produce a variety of metabolites. One of the major categories of phenolic compounds in walnuts are ellagitannins [64], which are hydrolyzed to produce ellagic acid and further acted upon by the gut microbiota to produce a series of metabolites known as urolithins [65]. Urolithins are better absorbed than ellagitannins and are thus transported to peripheral tissues or excreted through the urine [5,38,66]. Studies have shown urolithins to be valid biomarkers of walnut consumption [63,67,68] with higher concentrations of the metabolite found 12 h or longer following walnut ingestion [69].

Our study has many strengths. The WAHA study has a significantly long duration of intervention (2 years). The study also includes a relatively large number of participants who demonstrated excellent compliance, with a retention rate of 90%. Moreover, the dietary intake data were extensive, having been acquired using multiple 24-h recalls (up to five recalls) obtained throughout the 2 year period and carefully matched with Phenol-Explorer values to obtain a profile of its polyphenol content. 

The main limitation was that the urine samples were obtained from fasting participants following the first morning void, and as such, may not have captured a large enough quantity and diversity of phenolic metabolites excreted in a 24-h urine or in a longer collection period. It is well-known that the half-life of most polyphenol metabolites is relatively short and typically appear in urine within 1 to 24 h following ingestion [36]. Some phenol metabolites produced by microbiota such as urolithins may not be detected or quantified by the F–C reagent assay used in this study. A potential limitation is that despite an open recruitment policy, our study participants included a higher proportion of females than males. Additionally, the diets of the participants in the walnut group showed a higher mean energy and fat intake than the habitual diet group, which was partially mitigated through energy adjustment.

## 5. Conclusions

A single food such as walnuts eaten daily can increase dietary polyphenol intake. This is important as we now know that polyphenols have significant health benefits, being powerful anti-inflammatory and antioxidant phytochemicals. To reduce the risk for age-related chronic diseases, it may be prudent to include nuts such as walnuts as part of the usual diet to not only benefit from the unsaturated fatty acids and other nutrients that have CVD and neuroprotective effects, but also increase the polyphenol intake, which can synergistically influence the disease risk in a favorable manner.

## Figures and Tables

**Figure 1 nutrients-15-01253-f001:**
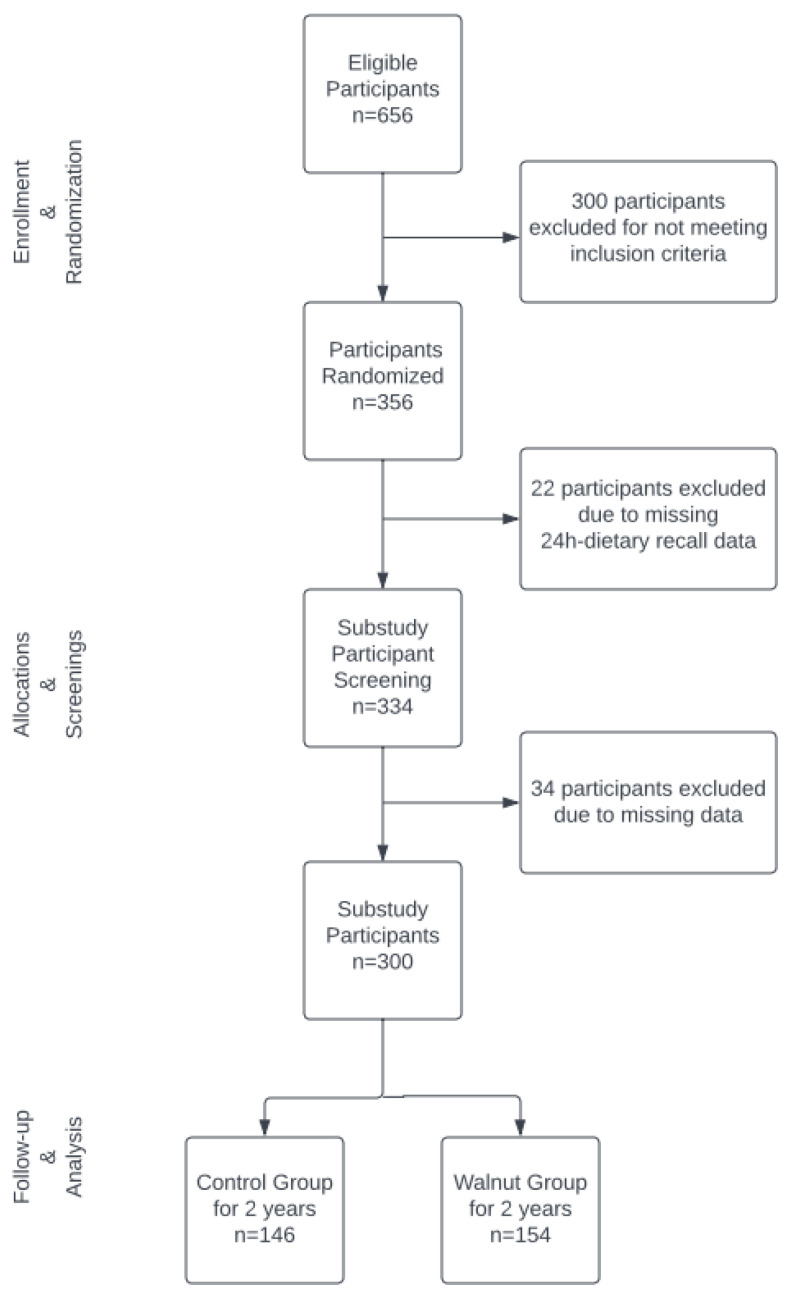
Participant flowchart at LLU site.

**Table 1 nutrients-15-01253-t001:** The subject characteristics of the study by treatment group (*n* = 300).

Baseline Characteristics		Control	Walnut	*p*-Value ^2^
		*n* = 146	*n* = 154	
Sex, *n* (%)	Women	99 (67.8)	97 (63.0)	0.450
	Men	47 (32.2)	57 (37.0)	
Race, *n* (%)	White	111 (76.0)	121 (78.6)	0.698
	Non-White	35 (24.0)	33 (21.4)	
Age, years, mean (SD) ^1^		69.42 (3.64)	70.08 (4.04)	0.141
BMI, kg/m^2^, mean (SD) ^1^		27.65 (4.92)	27.47 (5.02)	0.744
Education, *n* (%)	<12 years	19 (13.0)	14 (9.1)	0.368
	>12 years	127 (87.0)	140 (90.9)	
Ever smoked, *n* (%)	Never	143 (97.9)	147 (95.5)	0.379
	Ever	3 (2.1)	7 (4.5)	
Waist circumference, cm mean (SD) ^1^		98.08 (12.45)	98.86 (14.50)	0.676
Hip circumference, cm. mean (SD) ^1^		106.81 (11.02)	105.63 (10.87)	0.356

^1^ Two sample *t*-test was used to calculate the means and standard deviations. ^2^ Mann-Whitney tests were used for comparisons between treatments.

**Table 2 nutrients-15-01253-t002:** Dietary intake of macronutrients per day by treatment group.

Variables	Control	Walnut	*p*-Value ^1^
	*n* = 146	*n* = 154	
Energy, kcal, mean (SD)	1608 (453)	1836 (536)	<0.001
Total carbohydrate, g, mean (SD)	194 (65)	207 (80)	0.143
Total dietary fiber, g, mean (SD)	21 (8)	25 (11)	<0.001
Total fat, g, mean (SD)	63 (22)	84 (26)	<0.001
Saturated fatty acids, g, mean (SD)	21 (10)	22 (10)	0.347
Monounsaturated fatty acids, g, mean (SD)	22 (9)	25 (9)	0.012

^1^ Two sample *t*-test was used to calculate the mean and SD.

**Table 3 nutrients-15-01253-t003:** Daily intake of energy-adjusted dietary polyphenols ^1^ by treatment group.

Variables	Control	Walnut	*p*-Value ^7^
	*n* = 146	*n* = 154	
Total polyphenols ^2^, mg, median [IQR] ^8^	1897 [1369, 2496]	2479.99 [1956, 3146]	<0.001
Total flavonoids ^3^ (flavones, flavonols, and anthocyanidins), mg, median [IQR]	28.8 [15.4, 54.4]	56.1 [41.7, 83.9]	<0.001
Flavanols, mg ^4^, median [IQR]	139.6 [60.7, 277.3]	174.2 [89.8, 298.4]	0.036
Phenolic acids ^5^, mg, median [IQR]	242.2 [88.8, 398.3]	367.8 [245.7, 569.2]	<0.001
Lignans ^6^, mg, median [IQR]	27.4 [13.9, 44.8]	24.1 [13.4, 44.2]	0.514

^1^ Polyphenol composition of food items obtained from Phenol-Explorer version 3.6. All phenolic values were energy adjusted. ^2^ Total polyphenols were measured using the Folin–Ciocalteu assay. ^3^ Total flavonoids (flavones and flavonols) were measured using chromatography after hydrolysis. ^4^ Flavanols (proanthocyanidins) were measured using normal phase HPLC. ^5^ Phenolic acids (ellagitannins/tannins) were measured using chromatography after hydrolysis. ^6^ Lignans were measured using chromatography after hydrolysis. ^7^ Mann–Whitney tests were used for comparisons between treatment. *p* < 0.05 indicates significance. ^8^ Interquartile range (IQR).

**Table 4 nutrients-15-01253-t004:** Contribution of food groups by treatment group to the mean daily intake in mg/d of the total polyphenols and polyphenol subclasses.

		Total Polyphenols		Flavonoids (Flavones, Flavonols, Anthocyanidins)		Flavonoids (Flavanols)		Phenolic Acids		Lignans	
		Control	Walnut	Control	Walnut	Control	Walnut	Control	Walnut	Control	Walnut
Food Groups		Mean (SD)									
Beverages ^1^		804 (674)	886 (778)	27 (72)	30 (81)	7 (16)	6 (15)	214 (229)	240 (305)	0.7 (2)	0.6 (2)
Fruits ^2^		351 (280)	440 (374) *	15 (23)	16 (22)	81 (104)	102 (134)	16 (25)	21 (26)	7 (12)	8 (11)
Nuts ^3^		40 (77)	632 (182) ***	0.4 (1)	26 (7) ***	9 (22)	27 (11) ***	5 (16)	178 (52) ***	2 (6)	1.5 (6)
Legumes ^4^		244 (470)	317 (734)	1.3 (5)	1.4 (3)	4 (28)	9 (62)	4 (8)	5 (10)	2 (4)	2 (5)
Vegetables ^5^		226 (191)	269 (307)	9 (8)	9 (8)	_	0.28 (2) *	4 (4)	5 (6)	22 (32)	26 (45)
Grains ^6^		124 (138)	130 (141)	1.1 (8)	0.34 (1)	20 (72)	16 (47)	30 (23)	31 (27)	1.5 (3)	1 (2)
Chocolate ^7^		86 (180)	114 (274)	-	-	43 (93)	59 (144)	0.7 (1.4)	1 (2.3)	0.04 (0.4)	0 (0.1)
Spices ^8^		22 (67)	31 (86)	0.02 (0.1)	0.1 (0.7)	-	-	0.01 (0.04)	0.01 (0.02)	0.05 (0.2)	0.03 (0.1)
Fats and oils ^9^		7 (6)	7(6)	0.2 (0.5)	0.2 (0.2)	0.01 (0.1)	0.01 (0.1)	0.01 (0.1)	0.04 (0.2)	0.2 (0.3)	0.2 (0.3)
Miscellaneous ^10^		51 (78)	60 (107)	0.5 (1.1)	0.7 (1.6)	24 (42)	32 (62)	0.24 (1)	0.23 (1)	-	-

* *p* < 0.05, *** *p* < 0.001. Data are the means and SDs of polyphenol intake by treatment group. ^1^ Beverages include vegetable juice, fruit juice, coffee, tea, alcoholic beverages, rice milk, and non-alcoholic beer. ^2^ Fruits include whole, canned, frozen, fried, and dried fruits and jams. ^3^ Nuts include walnuts and other tree nuts, raw and roasted nuts, mixed nuts, and peanuts. ^4^ Legumes include dried beans, peas, soy, and soy-based products. ^5^ Vegetables include fresh, frozen, fried, and canned vegetables. ^6^ Grains include whole and refined grains, breads, pasta, and cereals. ^7^ Chocolate includes cocoa powder, cocoa mixed products, chocolate frosting, pudding mix chocolate, chocolate sauce, baking chocolate, chocolate-based candy and cake, and chocolate bars and cookies. ^8^ Spices include ground, dried, or powder spices such as thyme, turmeric, cumin, nutmeg, curry powder, cinnamon, tarragon, paprika, marjoram, taco seasoning, Italian seasoning, black pepper, allspice, rosemary, sage, cloves, parsley, and sage. ^9^ Fats and oils include oils such as olive, canola, sunflower, safflower, corn, vegetable oil, margarine, butter, and dressings. ^10^ Miscellaneous foods (Misc.) consist of mixed foods and dishes that include dairy and dairy products.

**Table 5 nutrients-15-01253-t005:** Comparison of the morning spot urine polyphenol excretion between treatment groups ^1^.

	Urine Total Polyphenols (mg GAE ^2^/L) Mean (95% CI)		Walnut–Control		Urine Total Polyphenols (mg GAE/g Cr ^2^) Mean (95% CI)		Walnut–Control (mg/g Cr)	
Time	Walnut	Control	Beta Estimate (SE)	*p*-Value	Walnut	Control	Beta Estimate (SE)	*p*-Value
Baseline	285 (267, 302)	286 (268, 304)	−1.09 (16.4)	0.9305	305 (282, 328)	303 (290, 327)	1.63 (16.4)	0.9207
Year 1	302 (285, 319)	279 (261, 297)	23.01 (16.4)	0.0662	333 (310, 356)	308 (284, 332)	24.93 (16.4)	0.1281
Year 2	295 (278, 313)	283 (265, 301)	12.62 (16.3)	0.3126	355 (333, 378)	337 (313, 360)	18.50 (16.3)	0.2579

^1^ Linear regression mixed models fitted for both variables (mg GAE/L, mg GAE/g Cr) included treatment, time, treatment × time interaction, age, gender, and BMI as fixed-effects terms and participants as the random-effects term. ^2^ Abbreviations: GAE = gallic acid equivalents, Cr = creatinine. *p* < 0.05 indicates significance.

**Table 6 nutrients-15-01253-t006:** Association between the spot urine polyphenol excretion at year 2 and the dietary intake of polyphenols and subclasses ^1^.

	Urine Polyphenols (mg GAE ^2^/L)		Urine Polyphenols (mg GAE/g Cr)	
Polyphenol Variables	Beta Estimate (SE)	*p*-Value	Beta Estimate (SE)	*p*-Value
Log total dietary polyphenols (mg/d)	8.33 (12.90)	0.5191	12.79 (18.95)	0.5002
Log total flavonoids (mg/d)	−14.27 (6.61)	0.0316	−17.23 (9.76)	0.0785
Log flavanols) (mg/d)	4.70 (5.35)	0.3801	−0.29 (8.09)	0.9717
Log phenolic acids (mg/d)	−3.24 (7.00)	0.6441	−0.16 (10.31)	0.9874

^1^ A linear regression model was fitted for each combination of urine polyphenol (dependent variable) and log dietary polyphenol (independent variable) while adjusting for age, gender, BMI. ^2^ Abbreviations: GAE = gallic acid equivalents, Cr = creatinine. *p* < 0.05 indicates significance.

## Data Availability

Detailed data relating to the study are available upon request from the corresponding author.

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
