# Peer review of "Effect of Walnut Supplementation on Dietary Polyphenol Intake and Urinary Polyphenol Excretion in the Walnuts and Healthy Aging Study"

_nutrients, 2023, doi:10.3390/nu15051253_

Round 1

Reviewer 1 Report

The authors should add the range of walnuts (g) added to the diet to provide 15% of energy intake.

The question addressed by this manuscript is whether adding a small amount of daily walnuts at 15% of total energy intake affects dietary polyphenol intake and subsequent polyphenol excretion in the urine.  This is a significant contribution as previous long term combination walnut and olive oil supplementation resulted in improvements for heart risk factors.  This study measured different polyphenols in the control and walnut added diet and found a significant increase in all the individual classes of polyphenols when walnuts were added.  As mentioned by the authors a 24 hour urine rather than spot urines should have been done to provide a more accurate measurement of urinary polyphenols.  The conclusions were correctly done from the data and a comprehensive reference list was made.

Author Response

The question addressed by this manuscript is whether adding a small amount of daily walnuts at 15% of total energy intake affects dietary polyphenol intake and subsequent polyphenol excretion in the urine.  This is a significant contribution as previous long term combination walnut and olive oil supplementation resulted in improvements for heart risk factors.  This study measured different polyphenols in the control and walnut added diet and found a significant increase in all the individual classes of polyphenols when walnuts were added.  As mentioned by the authors a 24 hour urine rather than spot urines should have been done to provide a more accurate measurement of urinary polyphenols.  The conclusions were correctly done from the data and a comprehensive reference list was made.

Thank you for your comments and suggestions, they are very valuable and helpful. We were able to add a range of walnuts (g) (30-60 g/d) added to the diet to provide ~ 15% of energy intake.

Reviewer 2 Report

In the present work, the authors addressed an interesting topic. However, I do have some concerns that authors should elaborate on before the Manuscript is accepted for the publications.

Polyphenols and their excretion is hard to record by using spot urine samples. The authors noted this and outlined as the limitation of their study. Still, in my  opinion, they should mention why they did not use the 24 hours collection. 

Also, the authors should explain why more women were included in the study.

Finally, my biggest concern is regarding the difference in walnut and control diet in terms of total energy and total fat intake. My question is was there any attempt to match those 2 diets in terms of this. Since 15 % of the energy is derived from the walnuts in walnuts diet, I believe the portion size of other fatty foods and sources could have been reduced to accommodate this.

Taking into account all above said and the possible impact of mentioned "limitations" on study results, I think this work should be accepted after the minor revisions made by the authors. 

Author Response

In the present work, the authors addressed an interesting topic. However, I do have some concerns that authors should elaborate on before the Manuscript is accepted for the publications.

Polyphenols and their excretion is hard to record by using spot urine samples. The authors noted this and outlined as the limitation of their study. Still, in my  opinion, they should mention why they did not use the 24 hours collection. 

We want to thank you for your helpful suggestions and comments. To address your first comment regarding why we did not use the 24-hour collection: Our study is a secondary data analysis that was based on the original study design [1]. Our team utilized the samples previously collected and processed in our lab. Therefore, we did not have control over what method of urine collection should have been followed.

Also, the authors should explain why more women were included in the study.

To address your second comment to explain why more women were included in the study: Due to the nature of the original protocol of the study [1], we had minimal control over participants, and we utilized participants who were already enrolled and had completed the study within the LLU cohort.

Finally, my biggest concern is regarding the difference in walnut and control diet in terms of total energy and total fat intake. My question is was there any attempt to match those 2 diets in terms of this. Since 15 % of the energy is derived from the walnuts in walnuts diet, I believe the portion size of other fatty foods and sources could have been reduced to accommodate this.

Taking into account all above said and the possible impact of mentioned "limitations" on study results, I think this work should be accepted after the minor revisions made by the authors. 

To address the last comment regarding the difference in walnut and control diets in terms of total energy and total fat intake and be able to match the 2 diets: In our secondary data analysis, there was no attempt to match the two diets in terms of total energy and total fat intake. Our experience with nuts studies, specifically walnuts, is that participants adding walnuts to their habitual diet had no adverse effect on body weight or body composition [2]. Additionally, per our original study protocol, we did not control participants' diet, but instead, participants were instructed to add walnuts daily to their normal dietary habits. However, this is a limitation of our study; thus, we added the suggestions to our limitation section. A potential limitation is that despite an open recruitment policy, our study participants included a higher proportion of females than males. Also, the diets of participants in the walnut group showed a higher mean energy and fat intake than the habitual diet group, which was partially mitigated through energy adjustment.

  1. Rajaram, S., Valls-Pedret, C., Cofán, M., Sabaté, J., Serra-Mir, M., Pérez-Heras, A. M., Arechiga, A., Casaroli-Marano, R. P., Alforja, S., Sala-Vila, A., Doménech, M., Roth, I., Freitas-Simoes, T. M., Calvo, C., López-Illamola, A., Haddad, E., Bitok, E., Kazzi, N., Huey, L., Fan, J., … Ros, E. (2017). The Walnuts and Healthy Aging Study (WAHA): Protocol for a Nutritional Intervention Trial with Walnuts on Brain Aging. Frontiers in aging neuroscience8, 333. https://doi.org/10.3389/fnagi.2016.00333
  2. Bitok, E.; Rajaram, S.; Jaceldo-Siegl, K.; Oda, K.; Sala-Vila, A.; Serra-Mir, M.; Ros, E.; Sabaté, J. Effects of Long-Term Walnut Supplementation on Body Weight in Free-Living Elderly: Results of a Randomized Controlled Trial. Nutrients2018, 10, 1317. https://doi.org/10.3390/nu10091317